Effects of nanoplastics on the gut microbiota of Pacific white shrimp Litopenaeus vannamei

Zhu Chenxi 1
Li Yiming 2
Liu Guoxing 3 4
Abdullah Anisah Lee 1 anisah@usm.my
Jiang Qichen 3 4 qichenjiang@live.cn
1 Geography, School of Humanities, Universiti Sains Malaysia , Penang , Malaysia
2 Fishery Machinery and Instrument Research Institute, Chinese Academy of Fisheries Sciences , Shanghai , China
3 Freshwater Fisheries Research Institute of Jiangsu Province , Nanjing , China
4 Low-temperature Germplasm Bank of Important Economic Fish (Freshwater Fisheries Research Institute of Jiangsu Province) of Jiangsu Provincial Science and Technology Resources (Agricultural Germplasm Resources) Coordination Service Platform , Nanjing , China
Syed Mudasir Ahmad
Electronic publication date: 2024 Jan 4
Publication date: 2024
Volume: 12
Electronic Location ID: e16743
Received 2023 Oct 5; Accepted 2023 Dec 11
Copyright: © 2024 Zhu et al.
Copyright year: 2024
Copyright holder: Zhu et al.
License: This is an open access article distributed under the terms of the Creative Commons Attribution License, which permits unrestricted use, distribution, reproduction and adaptation in any medium and for any purpose provided that it is properly attributed. For attribution, the original author(s), title, publication source (PeerJ) and either DOI or URL of the article must be cited.
License URL: https://creativecommons.org/licenses/by/4.0/

Keywords: Nanoplastics, Litopenaeus vannamei, Gut microbes

Funding: “JBGS” Project of Seed Industry Revitalization in Jiangsu Province JBGS [2021] 123 Central Public-interest Scientific Institution Basal Research Fund, CAFS 2022XT01 and 2021JBFM21 Freshwater Fisheries Research Center, CAFS 2021JBFM21 China Scholarship Council 202310710002 Fundamental Research Grant “HUMAN EXPOSURE TO MICROPLASTICS THROUGH SEAFOOD CONSUMPTION” This work was supported by grants from the “JBGS” Project of Seed Industry Revitalization in Jiangsu Province (JBGS [2021] 123), Central Public-interest Scientific Institution Basal Research Fund, CAFS (NO. 2022XT01), the Central Public-Interest Scientific Institution Basal Research Fund, Freshwater Fisheries Research Center, CAFS (2021JBFM21), the Central Public-interest Scientific. Institution Basal Research Fund, CAFS (2020TD36), and the China Scholarship Council (No. 202310710002), Fundamental Research Grant “HUMAN EXPOSURE TO MICROPLASTICS THROUGH SEAFOOD CONSUMPTION”. The funders had no role in study design, data collection and analysis, decision to publish, or preparation of the manuscript.

==============================
Nanoplastics (NPs) are an abundant, long-lasting, and widespread type of environmental pollution that is of increasing concern because of the serious threats they might pose to ecosystems and species. Identifying the ecological effects of plastic pollution requires understanding the effects of NPs on aquatic organisms. Here, we used the Pacific white shrimp (Litopenaeus vannamei) as a model species to investigate whether ingestion of polystyrene NPs affects gut microbes and leads to metabolic changes in L. vannamei. The abundance of Proteobacteria increased and that of Bacteroidota decreased after NPs treatment. Specifically, Vibrio spp., photobacterium spp., Xanthomarina spp., and Acinetobacter spp. increased in abundance, whereas Sulfitobacter spp. and Pseudoalteromonas spp. decreased. Histological observations showed that L. vannamei exposed to NP displayed a significantly lower intestinal fold height and damaged intestinal structures compared with the control group. Exposure to NPs also stimulated alkaline phosphatase, lysozyme, and acid phosphatase activity, resulting in an immune response in L. vannamei. In addition, the content of triglycerides, total cholesterol, and glucose were significantly altered after NP exposure. These results provided significant ecotoxicological data that can be used to better understand the biological fate and effects of NPs in L. vannamei.

Introduction

Current global plastic manufacturing and use is unsustainable, highly durable, poorly recycled and poorly managed, plastic consumer products have seen a 20-fold increase in production over the past 50 years (Borrelle et al., 2020; Lau et al., 2020; Walker & Fequet, 2023). Globally, an estimated 9.2 billion metric tons (Mt) of plastics have been produced, with more than 6.9 billion Mt ending up in landfills (Geyer, Jambeck & Law, 2017). Plastic pollution and environmental accumulation are of global concern (Nguyen et al., 2023; Wang et al., 2020). Borrelle et al. (2020) estimated that 19–23 million Mt of plastic waste generated globally entered aquatic ecosystems in 2016. The ultra-small size and low density of microplastics (MPs) allow them to travel widely and long distances through ocean currents (Ballent et al., 2012; Eriksson et al., 2013). Once released into the environment, micro-and nanoplastics (MNPs) degrade slowly and accumulate in aquatic organisms, eventually entering the food chain and, thus, possibly being consumed by humans (Guerrera et al., 2021), emphasizing the importance of studying the aquatic bioavailability of MPs in seawater.

Once in the environment, large pieces of plastic (>25 mm) continue to degrade into smaller plastic debris (medium plastics, 5–25 mm) or smaller MPs (<5 mm) or NPs (<1 μm or 1,000 nm), which can pose a risk to the aquatic environment and human health (Gasperi et al., 2018). An increasing number of studies have examined the negative effects of MNP pollution on the aquatic environment, aquatic organisms, and humans (Han et al., 2022; Liu et al., 2019; Walker et al., 2022). For example, a 28-day chronic experiment with 75-nm NPs on Japanese marsh shrimp (Macrobrachium nipponense) revealed that exposure to NPs resulted in decreased survival and reduced antioxidant enzyme activity and related gene expression levels and affected lipid peroxidase immunase activity (Li et al., 2020). Wang et al. (2021) and Chae et al. (2019) treated L. vannamei with 5-μm MPs and 44-nm NPs, and then analyzed their 16s gut microbial sequence, finding that MPs induced oxidative stress and negatively impacted the gut microbiome. The composition of gut microorganisms is associated with healthy homeostasis in the host. A total of 75-nm NPs disrupted the gut microbial homeostasis of Eriocheir sinensis (Han et al., 2023) and Procambarus clarkii (Han et al., 2022). Exposure to NPs was found to increase the abundance of pathogens and decrease the abundance of beneficial bacteria. In summary, it is essential to investigate the variation of gut microorganisms in L. vannamei using 16s sequencing.

L. vannamei has an important role in the food chain in different aquatic systems, from rivers to oceans (Suman et al., 2020). Given its filter-feeding behavior, L. vannamei could be more exposed to contaminants compared with other nonfilter-feeding aquatic organisms. There are many studies of the ecotoxicological effects of MPs on L. vannamei, but most have focused on MPs (e.g., Duan et al., 2021; Hariharan et al., 2022). Therefore, this study investigated the effect of 75-nm PS-NPs on the intestinal microbiology of, and physiological homeostasis in L. vannamei. The study was based on the hypothesis that these PS-NPs enter prawn tissues through dietary exposure as a result of changing toxicokinetics and fluctuating particle characteristics, potentially influencing immune responses and leading to major alterations in gut microbiota. The specific objectives were to investigate the impacts of NPs on the intestinal tissues of L. vannamei and the toxic effects of PS-NPs on biochemical indicators related to intestinal health and glycolipid metabolism and intestinal microbiotal composition. The results of this study contribute to the development of shrimp breeding, provide a foundation for future evaluation of NPs toxicity, reduce monitory loss, and ensure shrimp growth. This research provided important ecotoxicological information for a more in-depth understanding of the biological fate and effects of NPs on crustaceans.

Materials and Methods

Culturing of Litopenaeus vannamei

Healthy L. vannamei averaging 3.2 ± 0.2 g in weight were sampled randomly from local shrimp pools in Nanjing, China. Each 10-L tanks with 15 shrimps, all shrimps were divided into four groups of three replicates each, totaling 180 shrimps. All shrimp appeared to be undamaged and healthy. They were acclimated for 4 weeks and fed Yongxin special feed (Ningbo Qiangpu Feed Co.) three times a day at 03:00, 13:00 and 20:00 h, with a daily feeding weight of 5% of body weight. Each tank was filled with filtered seawater with sufficient security for the shrimp and was continuously aerated (salinity 30 psu, temperature 26 °C ± 0.2 °C, pH 8.0 ± 0.1).

Nanoplastics

PS-NPs with a green fluorescence label were purchased from BaseLine Chromtech Research Centre (Tianjin, China). The 10-mL aqueous solution contained monodispersed PS-NPs that had a nominal diameter of 75 nm and a concentration of 10 mg/mL.

Experimental design and sample collection

After acclimatization, shrimp were divided into four groups: control (CK), NP5, NP10, and NP20. Each group comprised three 10-L tanks with 15 shrimp reared in each tank. CK tanks contained normal seawater without PS-NPs, whereas tanks in the NP5, NP10, and NP20 groups contained 5, 10 and 20 mg/L of PS-NPs, respectively. The appropriate concentration of 1/3 PS-NPs solution in each 10-liter tank is changed every 24 h to minimize shrimp stress during water changes.

After 96 h of exposure to 10 mg/L NPs, intestinal and hepatopancreatic tissues were collected from 15 shrimp from each of the three tanks per group, resulting in 45 shrimp intestinal and hepatopancreatic tissue samples, which were stored in liquid nitrogen in preparation for subsequent DNA extraction, biochemical index assays and histomorphometric analyses resulting in six replicate 16S rRNA sequences samples, three replicate biochemical indicator samples and three replicate histomorphometric analyses for CK and NP10 groups. In addition, hepatopancreas and intestines were extracted from total 45 shrimps from the three replicate tanks in the each NP5 and NP20 groups for measurement of three replicate biochemical parameters and three replicate histomorphometric analyses.

Histopathology

Shrimp were washed three times with sterile seawater and then placed in 4% paraformaldehyde for fixation and histopathological analysis. After 24 h of fixation, shrimp samples were dehydrated, cleaned with 75% alcohol, the trimmed wax blocks were sliced in a paraffin slicer to a thickness of 4 μm. Hemoglobin and eosin (H&E) was used to stain sections (Cardiff, Miller & Munn, 2014). In total, samples from each of the groups were used for histopathology.

Biochemical parameter measurements

Assay kits from the Nanjing Jiancheng Bioengineering Institute (Nanjing, China) were used to detect lysozyme (LZM: A050-1-1), acid phosphatase (ACP: A060-1-1), and alkaline phosphatase (AKP: A059-1-1) activity, and triglycerides (TG: A110-1-1), total cholesterol (TCHO: A111-1-1), and glucose (GLU: F006-1-1) concentrations, following the manufacturer’s instructions. After being stored at −80 °C, 0.2 g of shrimp hepatopancreas or intestine was added to 1.8 mL of saline solution before being homogenized at 4 °C or in an ice bath and used as a test solution for the assays indicated above.

Gut flora 16S rRNA sequences

Using a QIAamp DNA kit (Qiagen, Hilden, Germany), the total gut microbial DNA was extracted and purified from the gastrointestinal contents of each group of shrimp. The DNA concentration was then measured using agarose gel electrophoresis and a NanoDrop2000 spectrophotometer (Thermo Fisher Scientific, Waltham, MA, USA). Genomic DNA was used as a template, and PCR was performed using specific primers with barcodes and Tks Gflex DNA Polymerase (Takara Bio Inc., Shiga, Japan) to ensure amplification efficiency and accuracy. According to the target sequences of 16S rRNA-coding genes V3 and V4, PCR amplification was performed using the forward primer 343F: 5′-TACGGRAGGCAGCAG-3′ and the reverse primer 798R: 5′-AGGGTATCTAATCCT-3′. PCR amplification was performed according the manufacturer’s instructions. The PCR reaction consisted of one cycle of 94 °C for 5 min, 26 cycles of the three temperatures 94 °C for 30 s, 56 °C for 30 s, and 72 °C for 20 s, followed by one cycle of 72 °C, and final storage at 4 °C. We sent the samples to oebiotech, China for further high-throughput sequencing to obtain raw data. After downloading the data, the unprocessed data sequences were first trimmed using cut adapt software. The double-ended raw data extracted by the previous step were then subjected to quality filtering, noise reduction, splicing, and chimaera removal QC analysis using DADA2 (Callahan et al., 2016). This was done in accordance with QIIME 2 (Wu et al., 2015) to obtain sequences and amplicon sequence variant (ASV) abundance status.

Bioinformatics analysis

Using Qiime software, the observed-species, abundance-based coverage estimator (ACE), PD-whole-tree, Chao l, Shannon, and Simpson indices were calculated to assess the microbial community diversity, which was also analyzed using principal coordinate analysis (PCoA). Linear discriminant analysis (LDA) and effect size (EfSe) were used to reveal individual colonies, and the LDA threshold was established at results.

Data analysis

The mean and standard error (mean ± SE) of all data from this experiment were calculated using SPSS. One-way ANOVA, followed by Duncan’s multiple comparisons, were used to analyze group differences. P < 0.05 indicated significant differences, whereas P < 0.01 indicated highly significant differences. Graphs were generated using R and Graphpad Prism 10.

Results

The 16S rRNA sequencing data and species evaluation

Based on 16S rRNA sequencing data (Fig. 1A), CK samples clustered closely together, as did those of NP10, suggesting that there was a good grouping effect. This result suggests that further study of the microbial composition of the gut is highly desirable. Figure 1B shows the distribution of the diversity index in each group. Exposure to NPs did not have a significant impact on the variety of microbial species found in the gut (P > 0.05).

Figure 1 Comparative analysis of samples (A) PCoA analysis, (B) comparison of ASV diversity indices between groups violin plot.

NS, non-significant.

There was an increased abundance of Proteobacteria and decreased abundance of Bacteroidota after NPs treatment (Fig. 2A). Specifically, the abundance of Vibrio spp., Photobacterium spp., Xanthomarina spp., and Acinetobacter spp. increased and that of Sulfitobacter spp., Pseudoalteromonas spp., and Flavobacterium spp. decreased after NPs treatment. Vibrio spp. were most abundant, accounting for nearly 35% of species in all groups (Fig. 2B). The abundance of Pseudoalteromonas spp., Aliivibrio spp., and Flavobacterium spp. in the gut of L. vannamei reduced significantly after 10 mg/L NPs treatment (P < 0.05), whereas that of other genera did not change significantly (Fig. 3A). The changes in L. vannamei gut microorganisms at the species level after NPs treatment are shown in Fig. 3B. The aliivibrio_fischeri_g__Aliivibrio species of Aliivibrio spp. reduced significantly after 10 mg/L NPs treatment (P < 0.05).

Figure 2 Community structure of gut microbes of L. vannamei in the CK and NP10 treatment groups at the (A) phylum and (B) genus levels.

Figure 3 Differential abundance at the (A) genus and (B) species level of gut microbes of L. vannamei in the control and NP10 treatment groups.

The different colors represent the different groups of the samples, and the vertical coordinates indicate the relative abundance values of the species, dots near the bar graph represent unusual values.

Biochemical component analysis

Changes in the activity of LZM, ACP, and AKP and concentration of TG, TCHO, and GLU occurred after exposure of L. vannamei to NPs (Fig. 4). There was a statistically significant increase in the activity of LZM, ACP, and AKP in L. vannamei exposed to 20 mg/L NPs (P < 0.05) compared with the CK group. There was a statistically significant decrease in the content of TG, TCHO, and GLU in L. vannamei exposed to 20 mg/L NPs (P < 0.05) compared with the CK group.

Figure 4 Biochemical components change after different concentration NPs exposure to L. vannamei. NP5, NP10 and NP20 are the treatment groups, CK is control group.

Data are means ± SE. *P < 0.05; **P < 0.01.

Histomorphometric analysis of shrimp intestine

Histological observations showed that exposure of L. vannamei to NPs adversely affected the gut morphology. Compared with the CK group, the groups exposed to NPs had substantially lower intestinal fold height (portion of the arrow) and a degraded intestinal structure (Fig. 5). We randomly selected five intestinal fold heights in each group and measured and statistically analyzed the results in Table S7 and Fig. S1. The results showed that experimental fold height was significantly reduced in the NP10 and NP20 groups (P < 0.05).

Figure 5 Intestine morphology of Pacific white shrimp L. vannamei treated with different concentrations of NPs compared with control shrimp.

(A) CK group; (B) 5 mg/L, (C) 10 mg/L, and (D) 20 mg/L. Arrows indicate the intestinal fold height.

Discussion

Given their small size, NPs are easily absorbed by aquatic organisms, resulting in significant impacts on the gut microbial stability in these animals. NPs are being extensively explored as novel contaminants in aquatic animals (Han et al., 2022; Liao et al., 2022). The current study used 16s rRNA sequencing to investigate the effects of NPs on the gut microbes of L. vannamei. Although NPs exposure did not significantly affect the gut microbial species diversity of L. vannamei, PCoA revealed significant differences in the most abundant phyla after NPs exposure. These findings suggested that NPs exposure altered the gut floral composition of L. vannamei. After NPs treatment, the abundance of Proteobacteria increased, whereas that of Bacteroidota decreased, similar to results reported in zebrafish exposed to PS-NPs (Liao et al., 2022). Proteobacteria is typically the most prevalent phylum in aquaculture environments as a result of the use of bioflocs, biofilms, and recirculation systems (Huerta-Rábago et al., 2019; Liao et al., 2022; Martínez-Córdova et al., 2017). As a result, Proteobacteria dominate in all groups. Although Proteobacteria were the most abundant microbial phylum in the gut of L. vannamei, in agreement with previous research (Cardona et al., 2016; Elizondo-González et al., 2020; Zhang et al., 2014), this phylum includes many well-known pathogens (Amin et al., 2022; Wang et al., 2019). In contrast, Bacteroidota are primarily associated with glycogen metabolism, cholesterol metabolism, and bile acid synthesis (Xu et al., 2023). Thus, by decreasing the number of beneficial Bacteroidota and increasing the number of possibly pathogenic Proteobacteria, NPs could disturb the healthy balance of the microbiome in L. vannamei.

At the genus level, the abundance of Vibrio spp., Photobacterium spp., Xanthomarina spp., and Acinetobacter spp. increased and that of Flavobacterium spp., Sulfitobacter spp. and Pseudoalteromonas spp. decreased after NPs treatment. There was a significant reduction in Flavobacterium spp., a common pathogen of farmed fish that is capable of causing high mortality after exposure to NPs (Figueiredo et al., 2005). Unfortunately, little is known about Flavobacterium spp. in crustaceans and, thus, the impact of these bacteria on L. vannamei requires more investigation. Huang et al. (2020) first identified Acinetobacter venetianus as a potential bacterial pathogen of red leg disease in L. vannamei (Elizondo-González et al., 2020). Both this species and another potential bacterial pathogen Vibrio, were prominent in L. vannamei, accounting for nearly 35% of both CK and treatment groups. Under conditions of inadequate nutrition, low water quality, and immunosuppression, many Vibrio species are considered opportunistic pathogens of shrimp (Thompson, Iida & Swings, 2004). Vibrio has been linked to gastrointestinal disorders in cultivated shrimp, resulting in high mortality rates in shrimp aquaculture systems worldwide (Chen et al., 2017). These pathogens predominate in healthy shrimp, indicating that they might operate as opportunistic pathogens that cause disease in shrimp under appropriate environmental conditions. Thus, NPs exposure might destabilize the L. vannamei microbiome, promoting bacterial infection.

Histological observations showed that NPs exposure to L. vannamei adversely affected the gut morphology. The shrimp exposed to NPs had considerably shorter intestinal fold heights and a damaged intestinal structure compared with the CK group, regardless of NPs concentration. This suggested that when L. vannamei consumed NPs, the intestinal tract was damaged, enabling pathogenic bacteria to flourish and, thus, resulting in an altered microbiome. Largemouth bass Micropterus salmoides exposed to NPs showed signs of intestinal damage, including a decrease in the height and density of intestinal villi (Chen et al., 2022). Related studies showed that 0.2, 0.5, and 1.0 μg (g shrimp)−1 of fluorescent red polyethylene microspheres accumulated in all tissues of L. vannamei, with more severe damage in muscles, midgut glands, liver opancreas, and gill tissues (Hsieh et al., 2021).

LZM is a naturally occurring anti-infective agent that has bactericidal effects on fish and functions as a regulator of the complement system and phagocyte activation (Hou, Liu & Li, 2022). AKP is an important regulatory enzyme that is involved in basic functions in all organisms (Liu et al., 2018). ACP is a lysosomal enzyme that aids the immune response by assisting in the death and digestion of microbial pathogens (Fu et al., 2017). Exposure to 20 mg/L NPs stimulated LZM, ACP, and AKP activity, indicating an immune response in L. vannamei. In addition, the levels of TG, TCHO, and GLU were significantly altered after NPs exposure, suggesting that L. vannamei metabolism was also disturbed by NPs consumption. Previous studies have also reported the metabolism-disrupting effect of MPs in other aquatic organisms (Zhao et al., 2021; Zhu et al., 2022). The current study showed that NPs not only altered the composition of intestinal microbial species in L. vannamei, but also damaged the intestine and caused changes in metabolic and immune system-related proteins. Such important ecotoxicological information contributes to our current understanding of the biological fate and consequences of NPs in L. vannamei.

Conclusion

The study used 16s rRNA sequencing to investigate the effects of NPs on the microbiome of L. vannamei. Proteobacteria were more numerous and Bacteroidota were less numerous following NPs treatment. After exposure to NPs, the abundance of Sulfitobacter and Pseudoalteromonas decreased, whereas that of Vibrio, Photobacterium, Xanthomarina, and Acinetobacter increased. The gut morphology of L. vannamei exposed to NPs was negatively impacted, based on histological observations. Compared with the CK group, the NPs exposure groups had considerably shorter intestinal fold height and a damaged intestinal structure. Exposure to 20 mg/L NPs also stimulated LZM, ACP, and AKP activity, indicating an immune response in L. vannamei. In addition, the levels of TG, TCHO and GLU were significantly altered after 20 mg/L NPs exposure. In conclusion, the present study, using 16s sequencing and some biochemical parameters, found that NPs affected the gut microbial homeostasis of shrimp, with an increase in harmful bacteria and a decrease in beneficial bacteria, generating oxidative stress and metabolic disorders. This study complements the study on the toxic effects of NPs on aquatic organisms.

Supplemental Information

Supplemental Information 1 Methods and raw data for Figure 4 for the determination of the relevant enzyme activities and biochemical indices.

Click here for additional data file.

Additional Information and Declarations

Competing Interests

Author Contributions

Data Availability

The authors declare that they have no competing interests.

Chenxi Zhu conceived and designed the experiments, analyzed the data, prepared figures and/or tables, and approved the final draft.

Yiming Li performed the experiments, authored or reviewed drafts of the article, and approved the final draft.

Guoxing Liu analyzed the data, prepared figures and/or tables, and approved the final draft.

Anisah Lee Abdullah performed the experiments, authored or reviewed drafts of the article, and approved the final draft.

Qichen Jiang analyzed the data, authored or reviewed drafts of the article, and approved the final draft.

The following information was supplied regarding data availability:

The group sequences are available at Sequence Read Archive (SRA): SRR26159600-SRR26159615.

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
