# Peer review of "Effects of nanoplastics on the gut microbiota of Pacific white shrimp Litopenaeus vannamei"

_PeerJ, doi:10.7717/peerj.16743_

## Round 0.1 · original submission · Major Revisions

The study by Zhu et al. to investigate the impacts of NPs on the intestinal tissues of L. vannamei and the toxic effects of PS-NPs on biochemical indicators and intestinal microbiota. The study based on using sequence information of 16 rRNA and biochemical parameters and Histological observations for understanding the effect of Effects of nanoplastics on the gut microbiota is trivial. The effect of nanoplastic should be validated by Realtime expression as well as some toxicological experiments. The manuscript lack clear conclusion.

**Language Note:** The review process has identified that the English language must be improved. PeerJ can provide language editing services - please contact us at copyediting@peerj.com for pricing (be sure to provide your manuscript number and title). Alternatively, you should make your own arrangements to improve the language quality and provide details in your response letter. – PeerJ Staff

Reviewer 1 ·

Basic reporting

Zhu et al.'s study investigates the impact of nanoplastics on Pacific white shrimp, with a primary focus on understanding how these minuscule plastic particles influence the shrimp's gut, immune response, and gut microbiota. The English language used requires some editing while revising the manuscript. The authors have provided enough literature to support their results; however, the impact of the NP in aquaculture is not assessed properly. Authors need to add about the significance of their research for the shrimp industry in terms of monitory loss and health perspectives. The discussion section would be excellent if the authors could provide examples related to studies on shrimp rather than fish.

Experimental design

The aims and hypothesis are mentioned; however, some information is lacking regarding the experimental design and methods used.
Line 78: The water in each 10-L tank was replaced every 24 h and the PS-NPs are refilled. Frequent water changes at such intervals can induce stress in the shrimp, potentially leading to a false immune or stress response. Authors should provide specific details regarding the frequency and volume of these water changes.
In line 91, please specify the thickness of the sections used in the study.
In lines 80-82: "After 96 hours of exposure to 10 mg/L NPs…." Is there a particular reason for exclusively using NP10 for 16s sequencing?
Lines 83-97: These sentences lack clarity, making it difficult to discern the sample sizes used. Please provide clarification.
Line 87: The content is unclear; more context is needed.
Line 96: It would be helpful to enclose the manufacturer's name in parentheses when mentioning a specific product.
Line 98: Specify whether it is 4°C or -4°C for clarity.
Lines 101-102: Did you establish any cutoff values for the 260/280 and 260/230 ratios? These values are essential for assessing the quality of the extracted DNA.
In line 107, when referring to "manufacturer's instructions," briefly describe the steps involved in the PCR reaction.
In line 108: You did not mention the steps taken after the PCR reaction. Did you perform sequencing to obtain raw data? Please describe the sequencing methods used, or specify if the samples were sent to a sequencing company for analysis.

Validity of the findings

The data are lacking for some of the treatment groups (N5 and N20) for 16S. Authors need to write in the manuscript why these groups were not included.
The results sub-sections should be elucidated in more detail.
In Figure 5C, the arrow measures beyond the intestinal epithelium, while in Figure 5A, it measures from the epithelium. Furthermore, the length appears to be greater in NP10 compared to NP5.
In line 19, were similar effects observed in other replicates as well? Before drawing conclusions, authors should provide details in this section, and it would be appropriate to state, "similar results were observed in other replicates as well."

The discussion is overall well-written.

Reviewer 2 ·

Basic reporting

The topic is novel and very interesting.

But, in the introduction section, please add some sections about the effects of nanoplastics on gut microbiota, otherwise it is unclear to the readers that what is the current status of this topic.

Additionally, your results section is too simple and some formats need to be consistent. For example, line 128-129 are empty. Indent should be consistent too such as line 130 has no ident but line 135, 141, etc has it.

Experimental design

The methods are standard but replicates should be carefully demonstrated to guide readers for the robustness of your experimental results.

In method section 2.2, do you have characterization data of your nanoplastics?

In line 65, please mention how many replicates.

In figure 2, please mention your replicates and does the figure show mean or one sample?

In figure 3, the p-value is too small to see.

How many replicates in figure 5?

Validity of the findings

The impact may not be assessed. The implications of your findings are missing though the conclusions are described.

Reviewer 3 ·

Basic reporting

Review for Chenxi Zhu et al “Effects of nano plastics on the gut microbiota of Pacific white shrimp Litopenaeus vannamei”

Chenxi Zhu et al have worked on the effect of nanoplastics (NP) on Pacific white shrimp Litopenaeus vannamei. Exposure to NP had a significant effect on the gut microbes of the shrimp with an increase in proteobacteria while bacteroidota decreased. with specific genera such as Vibrio, Photobacterium, Xanthomarina, and Acinetobacter increasing in abundance. Conversely, Sulfitobacter and Pseudoalteromonas decreased. The altered gut microbiota may have ecological consequences for L. vannamei. They also did histology of the shrimp’s intestine and observed structural damages due to NP exposure. Metabolically, the study found alterations in triglycerides, total cholesterol, and glucose levels in response to NP exposure. These changes in metabolic indicators indicate that NPs can disrupt the metabolic balance of L. vannamei.

The manuscript has many typographical errors and needs significant improvements. However, the experiments are well executed with rigor.

My concerns-

Line 19 Abbreviate polystyrene NP (PS-NP) at all the places necessary in the
manuscript.

Line 31 Highly durable should be in smaller cases.

Line 53, "with than other nonfilter-feeding" seems to have a typographical error or
missing words. It should be rephrased for clarity, such as "compared with
other nonfilter-feeding."

Line 60 It would be helpful to specify which biochemical indicators and intestinal
microbiota the study is investigating. Be more specific to provide a clear
roadmap of the research objectives.

Line 66 You have mentioned 40 L tanks. How many tanks you have taken and roughly
how many shrimps were taken in them?

Line 67 Give the composition of Yongxin special feed.

Line 69 Give unit of salinity in parts per thousand or %

Line 98 Remove the negative sign before 4°C if it is a typographical error.

Line 104 Tks Gflex DNA Polymerase (Takara Bio, city, country) mention city and
country.
Line 113 Give the full form of the ACE
Line 116 Reduce space between at results.
Line 143 and in the amount of TG, TCHO, and GLU? add decrease here as per figure.
Line 164 Give space after although.
Line 188 Consider rephrasing the sentence “exhibit intestinal damage, as
evidenced by a decrease in the height and density of intestinal villi’
to
“Largemouth bass Micropterus salmoides exposed to NPs showed signs of intestinal damage, including a decrease in the height and density of intestinal villi”

Line 200 rephrase the sentence "added to current understanding" to "contributes to our current understanding."

Figure 5 Provide magnification for this figure.

Experimental design

no comment

Validity of the findings

no comment

---

## Round 0.2 · accepted · Accept

I am pleased to accept the manuscript for publication in PeerJ.

Reviewer 1 ·

Basic reporting

The authors have done an excellent job of addressing comments provided by reviewers.

Experimental design

The reviewers' comments about the experimental design have been addressed.

Validity of the findings

The validity of the findings is nicely explained.

Reviewer 2 ·

Basic reporting

After revision for my previous suggestions, it is now clear and unambiguous with professional article structures.

Experimental design

The replicates have been demonstrated clearly.

Validity of the findings

The implication and impacts have been assessed after revision.

Reviewer 3 ·

Basic reporting

I appreciate the authors for their prompt and thorough revisions addressing my earlier concerns. The modifications have significantly strengthened the manuscript, and I am now satisfied with the completeness and clarity of the content.

Experimental design

no comment

Validity of the findings

no comment

Additional comments

I appreciate the authors for their prompt and thorough revisions addressing my earlier concerns. The modifications have significantly strengthened the manuscript, and I am now satisfied with the completeness and clarity of the content.